# Infectious diseases, comorbidities and outcomes in hospitalized people who inject drugs (PWID)

**Jacqueline Lim**[1☉¤], **Sureka Pavalagantharajah**[2☉], **Chris P Verschoor**[3☉], **Eric Lentz**[2☉], **Mark Loeb**[4☉], **Mitchell Levine**[5☉], **Marek Smieja**[4☉], **Lawrence Mbuagbaw**[5☉], **Dale Kalina**[2☉], **Jean-Eric Tarride**[5,6☉], **Tim O'Shea**[2,7☉], **Anna Cvetkovic**[2☉], **Sarah van Gaalen**[8☉], **Aidan Reid Findlater**[2☉], **Robin Lennox**[9☉], **Carol Bassim**[2☉], **Cynthia Lokker**[5☉], **Elizabeth Alvarez**[5,6]*

1 Faculty of Health Sciences, McMaster University, Hamilton, Ontario, Canada, 2 Department of Medicine, McMaster University, Hamilton, Ontario, Canada, 3 Health Sciences North Research Institute, Sudbury, Ontario, Canada, 4 Pathology and Molecular Medicine, McMaster University, Hamilton, Ontario, Canada, 5 Department of Health Research Methods, Evidence and Impact, McMaster University, Hamilton, Ontario, Canada, 6 Centre for Health Economics and Policy Analysis (CHEPA), McMaster University, Hamilton, Ontario, Canada, 7 Population Health Research Institute, Hamilton, Ontario, Canada, 8 Department of Family Medicine, Queen's University, Kingston, Ontario, Canada, 9 Department of Family Medicine, McMaster University, Hamilton, Ontario, Canada

☉ These authors contributed equally to this work.
¤ Current address: Faculty of Medicine, University of Toronto, Toronto, Ontario, Canada
* alvare@mcmaster.ca

**Data Availability Statement:** There are legal restrictions to sharing a de-identified data set. The reasons are outlined here: The original data is held

## Abstract

Injection drug use poses a public health challenge. Clinical experience indicates that people who inject drugs (PWID) are hospitalized frequently for infectious diseases, but little is known about outcomes when admitted. Charts were identified from local hospitals between 2013–2018 using consultation lists and hospital record searches. Included individuals injected drugs in the past six months and presented with infection. Charts were accessed using the hospital information system, undergoing primary and secondary reviews using Research Electronic Data Capture (REDCap). The Wilcoxon rank-sum test was used for comparisons between outcome categories. Categorical data were summarized as count and frequency, and compared using Fisher's exact test. Of 240 individuals, 33% were admitted to the intensive care unit, 36% underwent surgery, 12% left against medical advice (AMA), and 9% died. Infectious diagnoses included bacteremia (31%), abscess (29%), endocarditis (29%), cellulitis (20%), sepsis (10%), osteomyelitis (9%), septic arthritis (8%), pneumonia (7%), discitis (2%), meningitis/encephalitis (2%), or other (7%). Sixty-six percent had stable housing and 60% had a family physician. Fifty-four percent of patient-initiated discharges were seen in the emergency department within 30 days and 29% were readmitted. PWID are at risk for infections. Understanding their healthcare trajectory is essential to improve their care.

by the respective hospitals, and data sharing agreements with the involved institutions that contributed data prohibit the authors from making the dataset publicly available. The full dataset creation plan is available from the authors upon request. For requests for de-identified and specified data access, contact the faculty in charge of the dataset at alvare@mcmaster.ca or dcampbel@stjosham.on.ca, or if you are interested in requesting hospital data, please contact the Hamilton Health Sciences Decision Support Team at DecisionSupport@hhsc.ca.

**Funding:** The author(s) received no specific funding for this work.

**Competing interests:** The authors have declared that no competing interests exist.

## Introduction

Injection drug use is a major public health concern in Canada [1]. In 2016, there were an estimated 171,900 people who inject drugs (PWID) in Canada, representing 0.7% of the population [2]. PWID are at high risk for infectious complications of drug use, including blood-borne viral (e.g. Human immunodeficiency virus (HIV), hepatitis B and C) and bacterial infections (e.g. staphylococci that cause acute complications such as bacteremia with sepsis and endocarditis) [3,4].

Clinical experience indicates that PWID are hospitalized and re-hospitalized frequently [5]. However, relatively little is known about the health trajectory of this population in that there are sparse data about how frequently PWID present to acute care and their longitudinal outcomes over time. The nature of their admissions is unclear when admitted to hospital for infectious complications of injection drug use. Risk for complications once PWID are hospitalized, such as death or admission to critical care, are ill-defined. Special concerns exist around PWID with infections and patient-initiated discharges, generally referred to as discharge against medical advice (AMA), and among those who leave with a peripherally inserted central catheter (PICC) line in place for the administration of long-term antibiotics or other therapies [6,7]. Patient-initiated discharges have been associated with increased readmission, morbidity and mortality, and healthcare costs in PWID [5]. Previous cohort studies have aimed to understand the relationship between specific infectious diseases (e.g., HIV, hepatitis C) and intravenous drug use, but these studies have not looked at all drug-related infections, of which may be more urgent and life-threatening than chronic diagnoses. Prior studies have not considered patients' healthcare trajectories after hospitalization for infection, nor their effects on the health system [8–10]. Treatments of infections related to injection drug use, such as HIV and hepatitis C and their complications, have also shifted from acute care to community-based care as better treatments for these have been developed, so there has been a shift in infectious diseases treated in hospitals [11]. Understanding the types of infectious diseases managed in acute care and outcomes of PWID represents the first step in addressing these concerns. Achieving such an understanding could lead to focused interventions to enhance continuity of care in this population.

One objective of the current investigation was to review hospital admissions of PWID with infections who access hospitals in Hamilton, Ontario, Canada to understand the types of infections, comorbidities, hospital service use and outcomes, including patient-initiated discharge and death, among this population. This project will serve as a first, foundational step to understand PWID by establishing the feasibility of cohort studies among this group, and subsequently allow the analysis of their longitudinal healthcare trajectories needed to improve their overall care.

## Methods

### Study design

A retrospective chart review was conducted to study PWID admitted to hospitals in Hamilton, Ontario between 2013 and 2018 and who were treated for infection. The study protocol was approved by the Hamilton integrated Research Ethics Board (HiREB #5556). Due to the nature of the retrospective review, the requirement for obtaining informed consent from participants was waived by the ethics board.

### Sample size, setting, and inclusion

Based on clinical experience and current literature, the expected proportion of patient-initiated discharge in this population is 25–30% [7]. A sample size of 246 participants was required

to allow for identifying between 30 and 50 events, with a 95% confidence interval and a 5% margin of error [12]. This number would allow to test for differences in moderately prevalent factors between patient-initiated and planned discharges. Deaths during hospital admission were also used as an outcome measure but were not used for sample size determination. Since identification of PWID with infection can be challenging to conduct using discharge diagnoses, participants were identified using the following methods:

1. The Infectious Diseases consult services in Hamilton maintain a daily patient list from two local hospitals, Juravinski Hospital and Hamilton General Hospital, and have been doing so since 2013. Both participating hospitals are tertiary centres located in the downtown area of Hamilton. Juravinski Hospital is a full-service general hospital with 228 beds and Hamilton General Hospital is a teaching hospital with 607 beds. This list was reviewed to identify PWID by either identifying self-reporting PWID who were admitted for drug related infections (e.g. cellulitis), or by identifying those who fit other criteria leading to the possibility that they might use injection drugs (e.g., endocarditis).

2. Participants were also identified by searching hospital records for relevant International Statistical Classification of Diseases and Related Health Problems (ICD) Codes using the following terms: drug use, substance use, substance use disorder, drug abuse, drug addiction, injection drug use, opioid dependence AND cellulitis, bacteremia, endocarditis, HIV, hepatitis B, hepatitis C and osteomyelitis.

For each individual patient, data were extracted for their first identified hospital admission. If there were multiple admissions, the earliest admission was included. Participants were included in the study if they were identified to have injected drugs within the 6 months leading up to their index hospital admission. Use of injection drugs in the 6 months prior to admission was chosen, as this time period allows for balancing of self-reported drug use (i.e., individuals may be more open to disclosing past use than current use) with the relationship between drug use and infections (i.e., the infection is more likely to be related to drug use if the events are closer in time) [1]. Participants were excluded if they had not injected drugs within the past 6 months, had only a remote history of injection drug use, were unidentifiable using the hospital information system (Meditech), or were not treated for any infection.

## Data collection, management, and storage

Patient files were accessed securely using Meditech. Data extractors inputted patient information into Research Electronic Data Capture (REDCap), a secure, web-based software platform [13]. Patient information was collected using two forms on REDCap. Primary reviewers extracted data and entered information into a study key, which included identifying information, such as the participant's name, provincial health insurance number, and full date of birth. The second form was used to record details of the index hospital admission. Following primary extraction, secondary reviews were conducted by an independent reviewer who re-assessed details to ensure accuracy and consistency. For discrepancies between primary and secondary reviews, the inconsistency was logged, and a third reviewer was consulted. To respect the confidentiality and anonymity of patients, there were no paper records kept of the data at any point during this study. Only data extractors had access to REDCap, and all activities were logged through REDCap.

Data collected for every unique participant included demographic information (name, health insurance number, gender, date of birth, housing status, primary care physician) and information on the index admission: admitting diagnosis, diagnosis for which infectious disease service was consulted, comorbidities, HIV status, hepatitis B and C status, length of stay,

events during the hospital stay (social work consult, ICU admission, surgery), discharge disposition (patient-initiated discharge, discharged home or to long-term care facility, death), and emergency room visit or readmission to hospital within 30 days after discharge. Consult notes, operative reports (if applicable), notes from social workers, discharge summaries, laboratory reports, and diagnostic codes for the index admission were used to abstract patient data for these queries. Comorbidities included those identified previously, as listed in the electronic record system, and those identified during the index hospitalization. Information about current substance use was also collected, if known; this information was extracted from consult notes or notes from the social workers, in which patients openly disclosed the substances being used leading up to the index admission. Results of urine toxicology tests were also referenced to obtain data on substances used, if available. Prescribed substances were not included for this query. While all substances can be prescribed medications, in this paper the use of these substances was collected in the context of IVDU and the formulation for use was not extracted as data. Information pertaining to whether or not a patient was receiving opioid agonist therapy was collected based on consult notes as well. Names and insurance numbers were collected to ensure these were unique individuals, but were anonymized prior to statistical analysis to protect confidentiality and reduce the risk of identifying participants.

## Statistical analysis

For data pertaining to demographics, infectious disease diagnoses, and events during hospital stay, continuous variables were summarized as the median and range, and categorical variables as the count and frequency. To evaluate potential risk factors of our primary outcomes, patient-initiated discharge and death, patients were stratified by either outcome and the distribution of the aforementioned variable compared by either Wilcoxon rank-sum test or Fisher's exact test; missing data was removed prior to these analyses, and $p < 0.05$ was considered statistically significant. All analyses were performed in the R environment (v3.6) [14].

## Data audit

An audit was conducted for a subset of 10 charts to ensure that patient information was captured accurately. This subset of charts was randomly selected. Survey responses for these charts on REDCap were compared against the patient charts on Meditech. The audit was conducted by three of the authors, who independently ensured the survey responses reflected the original dictations. Discrepancies subject to interpretation were noted and brought forth to the other auditors. Changes to survey responses were only made following discussion and approval from the three authors conducting the audit.

## Data sharing and availability

There are legal restrictions to sharing the de-identified data set. The original data is held by the respective hospitals, and data sharing agreements with the involved institutions that contributed data prohibit the authors from making the dataset publicly available. The full dataset creation plan is available from the authors upon request. For requests for de-identified and specified data access, contact the faculty in charge of the dataset at alvare@mcmaster.ca, or for hospital data requests, please contact the Hamilton Health Sciences Decision Support Team at DecisionSupport@hhsc.ca.

## Results

From the 280 unique charts initially retrieved, our final eligible sample consisted of 240 patients. Following the application of the above criteria, 27 individuals were excluded from 211 identified through the first search method (infectious diseases inpatient lists), and 13 individuals were excluded from the 69 identified through the second search method (using ICD codes) from 964 charts found through the searches. Individuals were excluded on the basis of their drug use as well as their admitting diagnosis upon inspection of their charts on the Meditech system. If there was evidence in the consult notes to suggest that a patient had not injected drugs in the 6 months prior to their index admission, or the list of admitting diagnoses did not include an infectious disease, individuals were considered ineligible to be included in our study.

Demographic data are summarized in Table 1. At the time of admission, the average age of the sample was 38 years and 55% were male. One hundred fifty-eight (66%) of the sample were stably housed and 143 (60%) had a primary care physician at the time of their hospitalization. During the index hospital admission, 161 (68%) individuals had hepatitis C, 10 (4%) had HIV, and 4 (2%) had hepatitis B. Primary infectious disease diagnoses included bacteremia (31%), abscess (29%), endocarditis (29%), cellulitis (20%), sepsis (10%), osteomyelitis (9%), septic arthritis (8%), pneumonia (7%), discitis (5%), meningitis/encephalitis (2%), and other (7%). Overall, the median length of stay was 11 days, during which 129 (54%) received a consultation from social work, 79 (33%) were admitted to the ICU, and 87 (36%) underwent surgery. Twenty-one (9%) of the sample died during hospitalization and 28 (12%) discharges were self-initiated during the index admission. Within 30 days following discharge, 70 (29%) were seen in the ER again and 32 (13%) were readmitted to the hospital. For patient-initiated discharges, 15 (54%) were seen in the ER and eight (29%) were readmitted within 30 days of leaving the hospital. One hundred forty-four (60%) individuals had a PICC line placed during the index hospital admission, and five (2.1%) patients self-initiated their discharge with a PICC line in place (data not shown). Significant differences were identified among patient-initiated discharges in regard to housing status, primary care physician status, comorbid HIV, readmissions within 30 days, and ER visits within 30 days.

A variety of comorbid conditions were noted during hospitalization, the most common being chronic pain (20%), depression (18%), anxiety/panic disorder (12%), and asthma (12%). The median number of comorbidities was one and ranged from zero to nine. For the full list of comorbidities, see Table 2. For those with abscesses (N = 70), the majority were located on the upper extremities (26%), spine (23%), or hip/pelvis (14%); the full range of sites are summarized in Table 3.

The most commonly used opioids among the sample were hydromorphone (35%), unspecified opioids (20%) and heroin (19%). Common non-opioid substances used by the sample included cocaine (40%), methamphetamines (21%), and alcohol (18%) (see Table 4 for full list of substances used). The median number of substances used was two. Eighty-one (34%) of individuals were on opioid agonist therapy prior to and at the time of hospitalization, of which, 68 (28%) were receiving methadone and 13 (5.4%) were receiving buprenorphine/naloxone.

## Discussion

### Main findings

Between 2013–2018, 240 individual PWID were admitted with infections at two academic hospitals in Hamilton, Ontario. Other hospitals in the area were not included in this study, as our

**Table 1. Demographics, infectious disease diagnoses and outcomes.**

| Demographics and comorbid infectious disease diagnoses § | Total | Patient-initiated discharge (Missing N = 9) | | Died (Missing N = 1) | |
|---|---|---|---|---|---|
| | (N = 240) | Yes (N = 28) | No (N = 203) | Yes (N = 21) | No (N = 218) |
| Age (years), median (range) | 38 (18–68) | 35 (25–50) | 38 (18–68)* | 42 (21–63) | 37 (18–68) |
| Sex [Female], n (%) | 109 (45%) | 14 (50%) | 89 (44%) | 8 (38%) | 101 (46%) |
| Housing status [Stable], n (%) | 158 (66%)β | 12 (43%) β | 140 (69%) *β | 14 (67%)β | 143 (65%)β |
| Primary care physician [Yes], n (%) | 143 (60%)Φ | 10 (36%) β | 130 (64%)*Φ | 11 (52%)Φ | 131 (60%)Φ |
| HIV, n (%) | 10 (4%) | 4 (14%) | 6 (3%) * | 1 (5%) | 9 (4%) |
| Hepatitis B, n (%) | 4 (2%) | 1 (4%) | 3 (1%) | 0 (0%) | 4 (2%) |
| Hepatitis C, n (%) | 161 (68%) | 19 (68%) | 136 (67%) | 12 (57%) | 149 (68%) |
| Infectious disease diagnoses, n (%) § | | | | | |
| Bacteremia | 75 (31%) | 6 (21%) | 67 (33%) | 8 (38%) | 67 (31%) |
| Abscess | 70 (29%) | 6 (21%) | 64 (32%) | 2 (10%) | 68 (31%) * |
| Endocarditis | 70 (29%) | 8 (29%) | 55 (27%) | 9 (43%) | 60 (28%) |
| Cellulitis | 49 (20%) | 7 (25%) | 41 (20%) | 1 (5%) | 48 (22%) |
| Sepsis | 24 (10%) | 0 (0%) | 23 (11%) | 9 (43%) | 15 (7%) *** |
| Osteomyelitis | 21 (9%) | 3 (11%) | 17 (8%) | 0 (0%) | 21 (10%) |
| Septic arthritis | 18 (8%) | 1 (4%) | 17 (8%) | 0 (0%) | 18 (8%) |
| Pneumonia | 17(7%) | 1 (4%) | 16 (8%) | 1 (5%) | 16 (7%) |
| Discitis | 11 (5%) | 1 (4%) | 10 (5%) | 0 (0%) | 11 (5%) |
| Meningitis/encephalitis | 5 (2%) | 1 (4%) | 4 (2%) | 1 (5%) | 4 (2%) |
| Other infectious disease diagnosis | 17 (7%) | 3 (11%) | 13 (6%) | 0 (0%) | 17 (8%) |
| Diagnosis count/patient, median (range) | 1 (1–5) | 1 (1–4) | 1 (1–5) | 1 (1–3) | 1 (1–5) |
| Events and outcomes § | | | | | |
| Length of stay (days), median (range) | 11 (0–229)Φ | 8 (0–70) | 11 (0–229)Φ | 8 (1–27)Φ | 11 (0–229)Φ |
| Social work consult, n (%) | 129 (54%) | 18 (64%) | 107 (53%) | 12 (57%) | 116 (53%) |
| ICU admission, n (%) | 79 (33%)Φ | 6 (21%) | 68 (33%) | 20 (95%) | 58 (27%)***Φ |
| Surgery, n (%) | 87 (36%)Φ | 9 (32%) | 72 (35%) | 7 (33%) | 79 (36%)Φ |
| Readmission (30 days), n (%) | 32 (13%)Φ | 8 (29%) | 23 (11%) *Φ | N/A | 32 (15%)Φ |
| ER Visit (30 days), n (%) | 70 (29%)Φ | 15 (54%) Φ | 53 (26%) ** Φ | N/A | 70 (32%)Φ |

§Statistics are for the dichotomous category in square brackets or the "yes" category if not explicitly stated.

Statistically significant differences between outcome strata are denoted by asterisks

*** p<0.001

** p<0.01

*p<0.05.

Φ Missing <5%; β Missing <10%.

N/A—not applicable.

target sample size was achieved through applying our search to the two included hospitals. It is expected that more unique PWID are admitted to all Hamilton hospitals for infections. Within our sample, 21 patients (9%) died. Sepsis and ICU admission were associated with higher rates of death. During the index hospital admission, 79 (33%) individuals were admitted to the ICU, and 87 (36%) underwent surgery.

Twelve percent (28 individuals) had a patient-initiated discharge; which was associated with increased rates of hospital readmission and ER visits within 30 days of discharge. The proportion of patient-initiated discharges in our study is low compared to previous findings that report up to 25–30% of patient-initiated discharges among PWID [5]. We also found that PWID who had a self-initiated discharge were more likely to have unstable housing, lack a

**Table 2. Comorbidities.**

| | Comorbidities* | | Total (N = 240)<br>n (%) |
|---|---|---|---|
| | **Mental health** | Depression | 42 (18%) |
| | | Anxiety / panic disorder | 28 (12%) |
| | | Bipolar disorder | 16 (7%) |
| | | Attention-deficit/hyperactivity disorder (ADHD) | 12 (5%) |
| | | Schizophrenia / psychosis / hallucinations | 10 (4%) |
| | | Post-traumatic stress disorder (PTSD) | 9 (4%) |
| | | Other mental health conditions | 13 (5%) |
| | **Chronic pain** | Chronic pain | 48 (20%) |
| | **Neurologic** | Seizure disorder | 7 (3%) |
| | | Brain injury | 6 (2%) |
| | | Stroke | 5 (2%) |
| | | Other neurologic conditions | 9 (4%) |
| | **Pulmonary / respiratory** | Asthma | 29 (12%) |
| | | Chronic obstructive pulmonary disease (COPD) | 13 (5%) |
| | | Other pulmonary respiratory conditions | 4 (2%) |
| | **Cardiovascular** | Hypertension | 22 (9%) |
| | | Thrombosis (e.g., deep vein thrombosis, pulmonary embolism) | 13 (5%) |
| | | Congestive heart failure | 9 (4%) |
| | | Valvular conditions | 7 (3%) |
| | | Arrhythmias | 6 (2%) |
| | | Dyslipidemia | 5 (2%) |
| | | Other cardiovascular conditions | 14 (6%) |
| | **Gastrointestinal** | Gastroesophageal reflux disease (GERD) | 12 (5%) |
| | | Ulcer | 8 (3%) |
| | | Other gastrointestinal conditions | 16 (7%) |
| | **Metabolic / endocrine** | Diabetes (type 1, type 2 or unspecified) | 11 (5%) |
| | | Other metabolic / endocrine conditions | 9 (4%) |
| | **Renal** | Renal failure or dysfunction | 5 (2%) |
| | | Other renal conditions | 5 (2%) |
| | **Musculoskeletal / rheumatologic** | Arthritis | 12 (5%) |
| | | Systemic inflammatory conditions | 5 (2%) |
| | | Other musculoskeletal / rheumatologic | 15 (6%) |
| | **Hematologic / oncologic** | Anemia | 12 (5%) |
| | | Other hematological or oncological conditions | 7 (3%) |
| | **Comorbidity count/patient, median (range)** | | 1 (0–11) |

*Could have more than one comorbidity.

primary care physician, and be living with HIV. The relationship between homelessness, decreased antiretroviral use, increased active drug use and types of drugs used, specifically shorter-acting drugs such as cocaine, have been described [15–17]. The frequency of social work consults were not found to be significantly different for those with patient-initiated discharge. There is a need for further exploration on how to best support PWID, and in particular, individuals experiencing homelessness while in the hospital [6,18]. Thirty-four percent of the sample were using OAT at the time of admission, which is relatively low compared to the 55–66% reported nationally [19].

**Table 3. Site of abscesses.**

| Site of Abscess* | Total (N = 70)<br>n (%) |
|---|:---:|
| **Upper extremity** | 18 (26%) |
| **Spine** | 16 (23%) |
| **Hip / pelvis** | 10 (14%) |
| **Lower extremity** | 8 (11%) |
| **Chest** | 8 (11%) |
| **Abdominal** | 6 (9%) |
| **Head/neck** | 5 (7%) |
| **Brain** | 4 (6%) |
| **Not specified** | 3 (4%) |

*Could have more than one site.

## Fit within the literature

Our findings are concordant with Canadian national statistics and other studies. National surveillance initiatives report that 66–68% of PWID have evidence of chronic or past hepatitis C infection [3,20]. The proportion of our sample with hepatitis C was 68%. Similar reports

**Table 4. Substances used and OAT (Opioid agonist therapy).**

| Substances* | Total (N = 240)<br>n (%) |
|---|:---:|
| **Opioid** | |
| Hydromorphone | 83 (35%) |
| Unspecified opioid | 49 (20%) |
| Heroin | 46 (19%) |
| Fentanyl | 16 (7%) |
| Oxycodone | 15 (6%) |
| Morphine | 13 (5%) |
| Codeine | 3 (1%) |
| Hydrocodone | 1 (0%) |
| **Non-Opioid Substances** | |
| Cocaine | 96 (40%) |
| Methamphetamines | 51 (21%) |
| Alcohol | 43 (18%) |
| Cannabis | 31 (13%) |
| Benzodiazepines | 9 (4%) |
| Other | 39 (16%) |
| **Unknown** | 46 (19%) |
| **Substance count/participant, median (range)** | 2 (1–8) |
| **Opioid agonist therapy (OAT)** | |
| **On OAT** | 81 (34%)[Φ] |
| **Type of OAT** | |
| **Methadone** | 68 (28%) |
| **Buprenorphine / Naloxone** | 13 (5.4%) |

*Could use more than one substance.

Φ Missing <5%.

estimate that HIV prevalence among PWID has been increasing in Canada. In 1997, the prevalence of HIV among PWID was 4.7%, but more recent reports indicate this to be closer to 10% [3,20,21]. In our sample from 2013–2018, 4% of individuals were HIV positive. However, as a cross-sectional study, increases in incidence during this time were not ascertained. Similarly, our findings surrounding substances used is comparable to the statistics reported in a needs assessment and feasibility study done for the Hamilton supervised injection site. Survey responses in this report indicated that the top five most commonly injected drugs among respondents included crystal meth, hydromorphone, cocaine, heroin, and morphine [1]. The most commonly injected opioids in our study were hydromorphone and heroin, and of non-opioid substances, cocaine and methamphetamines were most common. The alignment with national health reports and current survey statistics ensures credibility and indicates our sample is similar to Canadian PWID at large.

However, our investigation is unique in its aim and depth through collecting data on relevant factors and health outcomes of PWID. A related study by this group is looking at features of health programs and services in Canada for the prevention and management of infections in PWID, the findings of which reveal a gap in the literature; there are very few studies focusing on hospitalization among PWID [22]. This investigation thus offers detailed insight on a variety of indicators that are not typically explored in current research, such as infectious diseases and outcomes linked to acute care for PWID.

## Strengths and limitations

Our study used individual-level data. Using these records ensures accurate reflection of recent approaches to care of PWID in local hospitals. Additionally, data were extracted in duplicate to minimize errors. Any discrepancies between primary and secondary reviews were adjudicated by a third independent reviewer. As a final step, an audit of the data was conducted to ensure the data correctly reflected patient records. Implementing such thorough protocols help ensure the consistency and accuracy of data collection.

However, electronic medical record systems are not perfect. Although hospital records are typically detailed and coherent, some patients were missing initial consult and/or discharge reports and details regarding their index admission were thus recorded as "unknown" or "missing" responses during data collection. Another limitation is the method of identifying eligible participants. As the majority of study participants were selected by referencing ID consult lists, our sample consists of patients with more severe presentations. Thus, the generalizability and applicability of our findings may be limited to more severe cases among the PWID population and in non-urban centres. It is also important to note that cross-sectional investigations, such as the present study, are incomplete in their ability to capture an individual's changing health status and their recurring interactions with the healthcare system. For instance, diagnoses of HIV and hepatitis could be overlooked if patients were not tested for these viral infections during the index admission. Patients may self-initiate their discharge during subsequent admissions, implicating their outcomes and risk of re-hospitalization in a manner that was not captured initially. This limitation highlights the importance of future studies dedicated to longitudinal analyses of the outcomes and trajectories of PWID. Lastly, it is important to recognize the limitations with our methods of statistical analysis. With bivariate analyses, our investigation is limited in the number of covariates that can be included.

## Implications for policy and practice

Given the high rates of ICU admissions, deaths, patient-initiated discharges and other indicators, there is a need for hospital-level interventions to address the needs of PWID with

infectious complications of drug use. Knowledge of the types of infections and comorbidities that PWID commonly present with may inform planning at the hospital level for health services and resource allocation. Additionally, understanding the characteristics of patient-initiated discharge among PWID allows for the development of targeted approaches to prevent said outcome.

Social determinants of health and continuity of care are evidently important factors in the health of PWID. Rates of unstable housing and access to primary care were low in our study population, and both were associated with higher rates of patient-directed discharge. As such, initiatives to enhance primary care attachment and address housing insecurity for PWID after hospitalization should be explored. Access to low-barrier HIV services and treatment, as well as pre-exposure prophylaxis in the prevention of HIV, are important considerations for this subgroup of PWID [23,24].

## Implications for future research

This study demonstrated that there are sufficient numbers of PWID who experience infectious complications of drug use to develop a prospective cohort to further elucidate the healthcare trajectories and health services utilization of PWID. Given the limitations of a cross-sectional study design, longer term and preferably prospective analyses of hospital admissions and health care utilization among PWID are warranted to understand the healthcare trajectories of this population. Approaching future research with a wider scope may lead to a more comprehensive understanding of healthcare trajectories, such as social service use and health outcomes, and can inform practice with more acuity.

Another avenue for investigation entails the linking of the current data with administrative information to understand the outcomes and risk factors applicable to PWID, such as re-hospitalization, death, resource utilization, and the associated economic costs. Having a validated search string to identify PWID with infections from administrative databases would be helpful to identify a larger sample to further explore risk factors identified in this and previous studies.

## Conclusion

Injection drug use poses a significant public health challenge, as PWID are at high risk of serious infectious complications of drug use that require admission to hospital. Furthermore, we found that PWID patients frequently self-initiate their discharge from the hospital, which was associated with hospital readmission and ER visits. Social factors such as housing status and access to primary care were also related to patient-initiated discharge, and may thus implicate the healthcare trajectories of PWID. Ergo, it is important to focus on the types of infections PWID experience, their comorbidities, as well as health care service use and outcomes. This may help in the planning of hospital and integrated health and social services for this high-risk population.

## Acknowledgments

We wish to thank all the data extractors who helped review charts: Alice Lu, Jessica Jung, Edward Koo, Ravinder Sandhu, Michael Collins, E. Dimitra Bednar, Mylini Saposan, Quinten K. Clarke, David Beisel, and Eden Shaul.

## Author Contributions

**Conceptualization:** Mark Loeb, Mitchell Levine, Marek Smieja, Lawrence Mbuagbaw, Dale Kalina, Jean-Eric Tarride, Tim O'Shea, Anna Cvetkovic, Sarah van Gaalen, Aidan Reid Findlater, Robin Lennox, Carol Bassim, Cynthia Lokker, Elizabeth Alvarez.

**Data curation:** Jacqueline Lim, Sureka Pavalagantharajah, Eric Lentz.

**Formal analysis:** Jacqueline Lim, Sureka Pavalagantharajah, Chris P Verschoor, Elizabeth Alvarez.

**Investigation:** Jacqueline Lim, Sureka Pavalagantharajah, Elizabeth Alvarez.

**Methodology:** Jacqueline Lim, Sureka Pavalagantharajah, Chris P Verschoor, Eric Lentz, Tim O'Shea, Anna Cvetkovic, Elizabeth Alvarez.

**Project administration:** Jacqueline Lim, Sureka Pavalagantharajah, Tim O'Shea, Elizabeth Alvarez.

**Software:** Chris P Verschoor, Tim O'Shea, Anna Cvetkovic.

**Supervision:** Elizabeth Alvarez.

**Writing – original draft:** Jacqueline Lim, Chris P Verschoor, Elizabeth Alvarez.

**Writing – review & editing:** Jacqueline Lim, Sureka Pavalagantharajah, Chris P Verschoor, Eric Lentz, Mark Loeb, Mitchell Levine, Marek Smieja, Lawrence Mbuagbaw, Dale Kalina, Jean-Eric Tarride, Tim O'Shea, Anna Cvetkovic, Sarah van Gaalen, Aidan Reid Findlater, Robin Lennox, Carol Bassim, Cynthia Lokker, Elizabeth Alvarez.

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
