## [Decision Letter · Decision Letter 0]

9 Jul 2021

PONE-D-21-11155

Infectious diseases, comorbidities and outcomes in hospitalized people who inject drugs (PWID)

PLOS ONE

Dear Dr. Alvarez,

Thank you for submitting your manuscript to PLOS ONE. After careful consideration, we feel that it has merit but does not fully meet PLOS ONE’s publication criteria as it currently stands. Therefore, we invite you to submit a revised version of the manuscript that addresses the points raised during the review process.

We look forward to receiving your revised manuscript.

Kind regards,

Judith I Tsui

Academic Editor

PLOS ONE

Journal Requirements:

2. Please include in your Methods section (or in Supplementary Information files) the participating hospitals/institutions.

3. In ethics statement in the manuscript and in the online submission form, please provide additional information about the patient records/samples used in your retrospective study. Specifically, please ensure that you have discussed whether all data/samples were fully anonymized before you accessed them and/or whether the IRB or ethics committee waived the requirement for informed consent. If patients provided informed written consent to have data/samples from their medical records used in research, please include this information.

Reviewers' comments:

Reviewer's Responses to Questions

**Comments to the Author**

1. Is the manuscript technically sound, and do the data support the conclusions?

Reviewer #1: Yes

Reviewer #2: Partly

2. Has the statistical analysis been performed appropriately and rigorously? 

Reviewer #1: Yes

Reviewer #2: No

3. Have the authors made all data underlying the findings in their manuscript fully available?

Reviewer #1: No

Reviewer #2: No

4. Is the manuscript presented in an intelligible fashion and written in standard English?

Reviewer #1: Yes

Reviewer #2: Yes

5. Review Comments to the Author

Reviewer #1: Summary: This manuscript used hospital charts/records to abstract data from people who inject drugs and hospital diagnoses and outcomes. 240 individuals were included in the analysis from two hospitals in Hamilton, Ontario. One third were admitted to the ICU, 36% had surgery, 9% died and 12% left AMA. Bacteremia was the most common diagnosis. Many of those who left AMA were seen in the ER or re-admitted within 30 days. In general, the topic of health outcomes in PWID is important to try and improve hospital care. However, the paper lacks details in some areas to help conceptualize the findings.

Major comments:

1. A better understanding of the hospitals included in this manuscript would be helpful—all hospitals in Hamilton? What kind of hospitals, etc. it sounds like these are urban? Those who aren’t familiar with Hamilton and/or Ontaria would benefit from a better understanding of this.

2. Could you please describe the data audit in more detail? What happened if discrepancies were found? Who conducted the audit?

3. In general, there are many areas that I would recommend adding more detail either in the body of the text or in a supplement. What criteria were used to abstract patient data including: HIV status (based on labs, meds, diagnosis codes’]?); Hep C status (labs? Treated/untreated?); comorbid conditions? Were these included if documented in a note or need ICD9/10? A line in the discussion describes missing HIV or Hep C diagnoses if they were tested during the index admission—so was this the only way these diagnoses were captured? The index admission was used but where chart records examined prior to index admission?

4. The use of two reviewers, a third as a tie-breaker, and an audit adds rigor to the study.

Minor comments:

1. Could you clarify that those charts who were excluded in paragraph 1 of the results were because they didn’t mean eligibility criteria upon further inspection?

2. Substances used: is this inpatient? Or prior to admission? Hydromorphone is a prescription drug but is it being misused/abused/bought in these settings?

Reviewer #2: This manuscript is a descriptive, retrospective chart review study of PWID admitted to two hospitals with infections in a Hamilton, Ontario between 2013 and 2018. The authors report demographics, diagnoses, as well as healthcare utilization for this cohort and provide stratified results by patient-initiated and death-status. While the study nicely describes PWID/ID hospital care utilization, the manuscript lacks focus and the principal conclusions that patient directed discharges and readmissions are common have been demonstrated previously in national datasets. I think the introduction and discussion could better frame the primary hypotheses and contextualize the limited analyses performed. Additionally, the authors frame the manuscript as an initial baseline study to address longer term trajectories and improve care for PWID but do not describe any addiction interventions or withdrawal management approaches in the hospital which is essential and could be one of the strengths of retrospective chart review study like this (similar studies using publicly available databases lack detailed inpatient medication administration records or other interventions). Overall, the authors have done considerable work to gather this data yet lacks key information and requires additional focus. Below are several additional suggestions to strengthen the manuscript.

Major comments:

1. Abstract: “little is known about outcomes” – The manuscript would be better framed with more specific identification of the gap and the goals. Which outcomes? There’s a fairly robust literature on outcomes related to specific infectious diseases (HIV, HCV, endocarditis) though less about this larger collection of diagnoses.

2. Intro: Little is known about “health trajectory of this population.” I would define this term as it is important theoretically to the framing (over what time period, what outcomes).

3. Intro: P 4, line 85: The authors argue that by including all infections among PWID together this manuscript fills a gap in the literature. Each of these infections has distinct presentations and clinical courses, I think the manuscript would be stronger with a theoretical argument about why it is crucial to group these infections together.

4. P5 line 98: “This project will set the stage to further study healthcare trajectories of PWID with infections.” I’m not clear where this is going or what this means. I would suggest using this space in the manuscript instead to set up the analyses you are doing in this paper.

5. P5 Line 99: “The overall aim…” – I would delete or make this sentence much more focused. Based on the analyses later in the manuscript, it seems the goal of this manuscript is to understand characteristics of people who do and do not have patient initiated discharges but the introduction doesn’t really set this up.

6. P5, Line 107 --- This is pretty well documented in the literature in addition to clinical experience.

7. Methods: I would encourage the authors to describe a bit more about how injection drug use in the last 6 months determined/verified.

8. If patient-initiated discharges are indeed the focus of this paper, it would be valuable to report on medication for opioid use disorder provision in the hospital and/or withdrawal management strategy which has been shown to be linked with discharge. The authors report on medication for opioid use disorder provision prior to the hospitalization but not how the hospital approaches addiction treatment.

9. Additional analyses to consider: do people with serious deeper infections leave later in the hospitalization than people with less serious infections (cellulitis, abscesses vs endocarditis, bacteremia, etc).

10. Are the comorbidities baseline or identified during the hospitalization?

11. Page 17 Line 293-94 – would move sentence about HIV/HCV diagnosis to the limitations section.

12. It is interesting that 55% of the population was injected unspecified opioids or hydromorphone. Can the authors provide additional information about the local context?

13. Limitations: This is descriptive data with bivariate analyses and does not adjust for co-variates. The authors could consider a basic, simple logistic regression model looking at factors associated with patient initiated discharge or mortality, but will be limited in the number of covariates that can be included. Regardless, the limitations section should acknowledge the limitations of this kind of analysis.

6. PLOS authors have the option to publish the peer review history of their article (what does this mean?). If published, this will include your full peer review and any attached files.

Reviewer #1: No

Reviewer #2: No

---

## [Author Response · Author response to Decision Letter 0]

7 Oct 2021

Response: Thank you for outlining formatting requirements. We have reviewed the samples provided. Our title page and main manuscript have been revised to ensure compliance with formatting requirements as noted in the templates provided above. 

2. Please include in your Methods section (or in Supplementary Information files) the participating hospitals/institutions.

Response: Thank you. As Reviewer #1 indicated the same suggestion, our methods section has been expanded to include a brief description of the two participating hospitals. 

3. In ethics statement in the manuscript and in the online submission form, please provide additional information about the patient records/samples used in your retrospective study. Specifically, please ensure that you have discussed whether all data/samples were fully anonymized before you accessed them and/or whether the IRB or ethics committee waived the requirement for informed consent. If patients provided informed written consent to have data/samples from their medical records used in research, please include this information.

Response: Thank you for outlining these instructions. We have elaborated on the patient records used under our Methods section in our manuscript. We also specified that we used patient names and insurance numbers to prevent duplicate data entries. We anonymized the data set prior to statistical analyses. 

Response: Thank you for outlining these instructions. The team has agreed that we wish to preserve patient privacy where possible, as our data contains potentially identifying information (such as hospital sites and diagnoses) and considering the sensitivity and stigma surrounding injection drug use. Our protocol was approved by the Hamilton integrated Research Ethics Board. We have briefly summarized these sentiments and included contact information for the ethics board in our cover letter. 

Comments to the Author

Reviewer #1: Summary: This manuscript used hospital charts/records to abstract data from people who inject drugs and hospital diagnoses and outcomes. 240 individuals were included in the analysis from two hospitals in Hamilton, Ontario. One third were admitted to the ICU, 36% had surgery, 9% died and 12% left AMA. Bacteremia was the most common diagnosis. Many of those who left AMA were seen in the ER or re-admitted within 30 days. In general, the topic of health outcomes in PWID is important to try and improve hospital care. However, the paper lacks details in some areas to help conceptualize the findings.

Response: Thank you for taking the time to review our manuscript, we appreciate your consideration and thoughtful feedback. We have addressed each comment to ensure our paper includes the details as suggested. 

Major comments:

1. A better understanding of the hospitals included in this manuscript would be helpful—all hospitals in Hamilton? What kind of hospitals, etc. it sounds like these are urban? Those who aren’t familiar with Hamilton and/or Ontaria would benefit from a better understanding of this.

Response: Thank you for this suggestion, we agree that further detail on the participating hospitals would benefit readers outside of Ontario. We have therefore listed the participating hospitals and included a brief description of them in our methods section. 

2. Could you please describe the data audit in more detail? What happened if discrepancies were found? Who conducted the audit?

Response: Thank you for your comment, we see that more explanation of the audit process is warranted to fully capture our team’s commitment to accuracy in data collection. A more thorough explanation of the data audit has been added to the methods section accordingly.

3. In general, there are many areas that I would recommend adding more detail either in the body of the text or in a supplement. What criteria were used to abstract patient data including: HIV status (based on labs, meds, diagnosis codes’]?); Hep C status (labs? Treated/untreated?); comorbid conditions? Were these included if documented in a note or need ICD9/10? A line in the discussion describes missing HIV or Hep C diagnoses if they were tested during the index admission—so was this the only way these diagnoses were captured? The index admission was used but where chart records examined prior to index admission?

Response: Thank you for your suggestion, we agree that the additional detail to our data extraction and collection process is needed, as the criteria for abstraction was not clear. We have elaborated upon this step in our methods section to indicate how and where data was extracted from in Meditech. 

4. The use of two reviewers, a third as a tie-breaker, and an audit adds rigor to the study.

Response: Thank you, we appreciate this feedback. We have further expanded on our data audit process to ensure this rigor. 

Minor comments:

1. Could you clarify that those charts who were excluded in paragraph 1 of the results were because they didn’t mean eligibility criteria upon further inspection?

Response: Thank you for suggesting this clarification, we agree this suggestion will improve the cohesiveness of our paper. We have thus restated our exclusion criteria in the first paragraph of the results section. 

2. Substances used: is this inpatient? Or prior to admission? Hydromorphone is a prescription drug but is it being misused/abused/bought in these settings?

Response: Thank you for your comment, we agree this was unclear in our original submission. We have included a more detailed explanation of how information regarding used substances was obtained from the charts. 

Reviewer #2: This manuscript is a descriptive, retrospective chart review study of PWID admitted to two hospitals with infections in a Hamilton, Ontario between 2013 and 2018. The authors report demographics, diagnoses, as well as healthcare utilization for this cohort and provide stratified results by patient-initiated and death-status. While the study nicely describes PWID/ID hospital care utilization, the manuscript lacks focus and the principal conclusions that patient directed discharges and readmissions are common have been demonstrated previously in national datasets. I think the introduction and discussion could better frame the primary hypotheses and contextualize the limited analyses performed. Additionally, the authors frame the manuscript as an initial baseline study to address longer term trajectories and improve care for PWID but do not describe any addiction interventions or withdrawal management approaches in the hospital which is essential and could be one of the strengths of retrospective chart review study like this (similar studies using publicly available databases lack detailed inpatient medication administration records or other interventions). Overall, the authors have done considerable work to gather this data yet lacks key information and requires additional focus. Below are several additional suggestions to strengthen the manuscript.

Response: Thank you for taking the time to review our manuscript. We greatly appreciate your insightful feedback for our paper. We would like to clarify that our goal with this chart review was the establish the feasibility of studying injection drug use in Hamilton. Studies in the past have described general statistics of people who inject drugs, but have not yet considered the nature of hospital admissions for this group. This investigation is a first step to develop a better understanding this population for future studies to be conducted surrounding trajectories and interventions. 

Major comments:

1. Abstract: “little is known about outcomes” – The manuscript would be better framed with more specific identification of the gap and the goals. Which outcomes? There’s a fairly robust literature on outcomes related to specific infectious diseases (HIV, HCV, endocarditis) though less about this larger collection of diagnoses.

Response: Thank you for your comment, we have clarified this point in our abstract. 

2. Intro: Little is known about “health trajectory of this population.” I would define this term as it is important theoretically to the framing (over what time period, what outcomes).

Response: Thank you for your comment, we have clarified this point in our introduction. We agree that the term “health trajectories” may not be intuitive and warrants clarification. 

3. Intro: P 4, line 85: The authors argue that by including all infections among PWID together this manuscript fills a gap in the literature. Each of these infections has distinct presentations and clinical courses, I think the manuscript would be stronger with a theoretical argument about why it is crucial to group these infections together.

Response: Thank you for this consideration to help strengthen our paper. We have clarified the importance of considering all drug-related infections for this population accordingly. 

4. P5 line 98: “This project will set the stage to further study healthcare trajectories of PWID with infections.” I’m not clear where this is going or what this means. I would suggest using this space in the manuscript instead to set up the analyses you are doing in this paper.

Response: Thank you for your comment, we understand our original wording may not have been clear. We have clarified our aims for our study and ensured it flows with our analysis. 

5. P5 Line 99: “The overall aim…” – I would delete or make this sentence much more focused. Based on the analyses later in the manuscript, it seems the goal of this manuscript is to understand characteristics of people who do and do not have patient initiated discharges but the introduction doesn’t really set this up.

Response: Thank you for your suggestion, we acknowledge that this sentence is rather broad. We have modified the sentence for more specificity and incorporated the suggestion from the last comment to ensure our aims are clearer and more reflective of our work. 

6. P5, Line 107 --- This is pretty well documented in the literature in addition to clinical experience.

Response: Thank you for your comment. We have changed this sentence to reflect this. 

7. Methods: I would encourage the authors to describe a bit more about how injection drug use in the last 6 months determined/verified.

Response: Thank you for this excellent suggestion. Reviewer #1 expressed a similar sentiment above. As such, we have added more detail to explain our data extraction process by including how each survey query was answered and which parts of the patient chart were referenced for each query. 

8. If patient-initiated discharges are indeed the focus of this paper, it would be valuable to report on medication for opioid use disorder provision in the hospital and/or withdrawal management strategy which has been shown to be linked with discharge. The authors report on medication for opioid use disorder provision prior to the hospitalization but not how the hospital approaches addiction treatment.

Response: Thank you for your comments. We would like to clarify that patient-initiated discharges represent one of the multiple outcomes we focused on in our investigation. Unfortunately, there is little available on how the participating hospitals respond to and address addictions treatment, aside from what was found in consult notes surrounding opioid substitution therapy. We agree that linking substitution or withdrawal management with discharge data would be an excellent addition. We hope that our study, as a first step into investigating outcomes of PWID, will be conducive to such analysis in the near future. 

9. Additional analyses to consider: do people with serious deeper infections leave later in the hospitalization than people with less serious infections (cellulitis, abscesses vs endocarditis, bacteremia, etc).

Response: Thank you for these excellent suggestions. We agree that such analyses would yield important and insightful considerations in developing strategies to better care for this population. As mentioned above, we hope our study will allow further analyses to be carried out among this population to better understand the impact of infections of varying severities. 

10. Are the comorbidities baseline or identified during the hospitalization?

Response: Thank you for this question. We have clarified that information on comorbidities included those at baseline (made available in the electronic records system) as well as those identified during the hospitalization.

11. Page 17 Line 293-94 – would move sentence about HIV/HCV diagnosis to the limitations section.

Response: Thank you for this suggestion, we agree that this makes more sense in structuring our paper. This portion originally in the fit within literature subsection has been moved above to the limitations subsection as suggested. 

12. It is interesting that 55% of the population was injected unspecified opioids or hydromorphone. Can the authors provide additional information about the local context?

Response: Thank you for suggesting this addition. We agree that providing additional information with local data would fortify the clinical picture. We have included local data from a needs assessment and feasibility study for the Hamilton supervised injection site to help contextualize our findings. This addition was written into our discussion section, under the fit within literature subsection. 

13. Limitations: This is descriptive data with bivariate analyses and does not adjust for co-variates. The authors could consider a basic, simple logistic regression model looking at factors associated with patient initiated discharge or mortality, but will be limited in the number of covariates that can be included. Regardless, the limitations section should acknowledge the limitations of this kind of analysis.

Response: Thank you for your comment. We agree that limitations in our statistical analyses should be recognized. We have addressed this issue in our limitations subsection.

---

## [Decision Letter · Decision Letter 1]

6 Dec 2021

PONE-D-21-11155R1Infectious diseases, comorbidities and outcomes in hospitalized people who inject drugs (PWID)PLOS ONE

Dear Dr. Alvarez,

Thank you for submitting your manuscript to PLOS ONE. After careful consideration, we feel that it has merit but does not fully meet PLOS ONE’s publication criteria as it currently stands. Therefore, we invite you to submit a revised version of the manuscript that addresses the points raised during the review process.

Specifically, please address reviewers' remaining concerns, and particularly specify if data underlying the findings is fully available.

We look forward to receiving your revised manuscript.

Kind regards,

Jianhong Zhou

Associate Editor

PLOS ONE

Journal Requirements:

Reviewers' comments:

Reviewer's Responses to Questions

**Comments to the Author**

1. If the authors have adequately addressed your comments raised in a previous round of review and you feel that this manuscript is now acceptable for publication, you may indicate that here to bypass the “Comments to the Author” section, enter your conflict of interest statement in the “Confidential to Editor” section, and submit your "Accept" recommendation.

Reviewer #1: All comments have been addressed

Reviewer #2: All comments have been addressed

2. Is the manuscript technically sound, and do the data support the conclusions?

Reviewer #1: Yes

Reviewer #2: Yes

3. Has the statistical analysis been performed appropriately and rigorously? 

Reviewer #1: Yes

Reviewer #2: Yes

4. Have the authors made all data underlying the findings in their manuscript fully available?

Reviewer #1: No

Reviewer #2: No

5. Is the manuscript presented in an intelligible fashion and written in standard English?

Reviewer #1: Yes

Reviewer #2: Yes

6. Review Comments to the Author

Reviewer #1: Thank you for responding to the reviewer comments. I think this has strengthened the manuscript. This manuscript contributes to the literature of PWID and hospitalizations, patient-initiated discharges, and deaths.

One minor comment--hydromorphone was the most commonly injected substance, though it is also a prescription pain medication. perhaps the first time that hydromorphone is introduced in the paper that it can be described as mis-used/illicit hydromorphone? I suspect that this is hydromorphone pills that are crushed and injected as opposed to the IV hydromorphone formulation but perhaps authors can comment on this? I think there may be differences in infection risk between these two formulations...

Reviewer #2: The authors have largely addressed my comments but I think the methods section still feels a bit disjointed and does not set up the specific analyses performed. Perhaps the authors could move their discussion of sample size as it related to number of events to the statistical analysis section. The authors report detailed data collection strategy but could lay out more clearly which outcomes they will examine. The statistical analysis section can then more clearly state which comparison will be performed, etc.

7. PLOS authors have the option to publish the peer review history of their article (what does this mean?). If published, this will include your full peer review and any attached files.

Reviewer #1: No

Reviewer #2: No

---

## [Author Response · Author response to Decision Letter 1]

20 Dec 2021

Please see below responses to reviewers’ comments:

Specifically, please address reviewers' remaining concerns, and particularly specify if data underlying the findings is fully available.

RESPONSE: A data sharing and availability section was added under methods

Journal Requirements:

RESPONSE: A review of referenced articles has been completed. No articles were found to have been retracted. A couple of links have been updated.

Reviewers' comments:

Reviewer's Responses to Questions

Comments to the Author

1. If the authors have adequately addressed your comments raised in a previous round of review and you feel that this manuscript is now acceptable for publication, you may indicate that here to bypass the “Comments to the Author” section, enter your conflict of interest statement in the “Confidential to Editor” section, and submit your "Accept" recommendation.

Reviewer #1: All comments have been addressed

Reviewer #2: All comments have been addressed

RESPONSE: Thank you

2. Is the manuscript technically sound, and do the data support the conclusions?

Reviewer #1: Yes

Reviewer #2: Yes

3. Has the statistical analysis been performed appropriately and rigorously?

Reviewer #1: Yes

Reviewer #2: Yes

4. Have the authors made all data underlying the findings in their manuscript fully available?

Reviewer #1: No

Reviewer #2: No

RESPONSE: A data sharing and availability section was added under methods

5. Is the manuscript presented in an intelligible fashion and written in standard English?

Reviewer #1: Yes

Reviewer #2: Yes

6. Review Comments to the Author

Reviewer #1: Thank you for responding to the reviewer comments. I think this has strengthened the manuscript. This manuscript contributes to the literature of PWID and hospitalizations, patient-initiated discharges, and deaths.

RESPONSE: Thank you!

One minor comment--hydromorphone was the most commonly injected substance, though it is also a prescription pain medication. perhaps the first time that hydromorphone is introduced in the paper that it can be described as mis-used/illicit hydromorphone? I suspect that this is hydromorphone pills that are crushed and injected as opposed to the IV hydromorphone formulation but perhaps authors can comment on this? I think there may be differences in infection risk between these two formulations...

RESPONSE: Thank you, this was clarified in the methods section under data collection.

Reviewer #2: The authors have largely addressed my comments but I think the methods section still feels a bit disjointed and does not set up the specific analyses performed. Perhaps the authors could move their discussion of sample size as it related to number of events to the statistical analysis section. The authors report detailed data collection strategy but could lay out more clearly which outcomes they will examine. The statistical analysis section can then more clearly state which comparison will be performed, etc.

RESPONSE: Clarifications were added throughout the methods section to address these points.

7. PLOS authors have the option to publish the peer review history of their article (what does this mean?). If published, this will include your full peer review and any attached files.

Do you want your identity to be public for this peer review? For information about this choice, including consent withdrawal, please see our Privacy Policy.

Reviewer #1: No

Reviewer #2: No

Thank you again for your reviews.

---

## [Decision Letter · Decision Letter 2]

25 Mar 2022

Infectious diseases, comorbidities and outcomes in hospitalized people who inject drugs (PWID)

PONE-D-21-11155R2

Dear Dr. Alvarez, 

We’re pleased to inform you that your manuscript has been judged scientifically suitable for publication and will be formally accepted for publication once it meets all outstanding technical requirements.

Kind regards,

Yu Mon Saw

Academic Editor

PLOS ONE

Additional Editor Comments (optional):

Reviewers' comments:

Reviewer's Responses to Questions

**Comments to the Author**

1. If the authors have adequately addressed your comments raised in a previous round of review and you feel that this manuscript is now acceptable for publication, you may indicate that here to bypass the “Comments to the Author” section, enter your conflict of interest statement in the “Confidential to Editor” section, and submit your "Accept" recommendation.

Reviewer #1: All comments have been addressed

2. Is the manuscript technically sound, and do the data support the conclusions?

Reviewer #1: Yes

3. Has the statistical analysis been performed appropriately and rigorously? 

Reviewer #1: Yes

4. Have the authors made all data underlying the findings in their manuscript fully available?

Reviewer #1: No

5. Is the manuscript presented in an intelligible fashion and written in standard English?

Reviewer #1: Yes

6. Review Comments to the Author

Reviewer #1: Thank you for addressing my comment re: formulations of medications. This helps to clarify that substances can be regularly prescribed but in this case were used illicitly and injected in this way.

7. PLOS authors have the option to publish the peer review history of their article (what does this mean?). If published, this will include your full peer review and any attached files.

Reviewer #1: No

---

## [Editor Report · Acceptance letter]

8 Apr 2022

PONE-D-21-11155R2 

Infectious diseases, comorbidities and outcomes in hospitalized people who inject drugs (PWID) 

Dear Dr. Alvarez:

I'm pleased to inform you that your manuscript has been deemed suitable for publication in PLOS ONE. Congratulations! Your manuscript is now with our production department. 

Kind regards, 

on behalf of

Dr. Yu Mon Saw 

Academic Editor

PLOS ONE